# Stroke and Etiopathogenesis: What Is Known?

**DOI:** 10.3390/genes13060978

**Published:** 2022-05-30

**Authors:** Tiziana Ciarambino, Pietro Crispino, Erika Mastrolorenzo, Antonello Viceconti, Mauro Giordano

**Affiliations:** 1Internal Medicine Department, Hospital of Marcianise, ASL Caserta, 81031 Caserta, Italy; 2Emergency Department, Hospital of Latina, ASL Latina, 04100 Latina, Italy; picrispino@libero.it; 3Distretto Sanitario Lauria, ASP Basilicata, 85044 Lauria, Italy; erika.mastrolorenzo@libero.it; 4Emergency Unit, San Giovanni Hospital, AOR San Carlo, 85042 Lagonegro, Italy; a.viceconti@libero.it; 5Department of Advanced Medical and Surgical Sciences, University of Campania “L. Vanvitelli”, 80138 Naples, Italy

**Keywords:** stroke, causes, etiopathogenesis, genetic, epigenetic, ischemic stroke

## Abstract

Background: A substantial portion of stroke risk remains unexplained, and a contribution from genetic factors is supported by recent findings. In most cases, genetic risk factors contribute to stroke risk as part of a multifactorial predisposition. A major challenge in identifying the genetic determinants of stroke is fully understanding the complexity of the phenotype. Aims: Our narrative review is needed to improve our understanding of the biological pathways underlying the disease and, through this understanding, to accelerate the identification of new drug targets. Methods: We report, the research in the literature until February 2022 in this narrative review. The keywords are stroke, causes, etiopathogenesis, genetic, epigenetic, ischemic stroke. Results: While better risk prediction also remains a long-term goal, its implementation is still complex given the small effect-size of genetic risk variants. Some authors encourage the use of stroke genetic panels for stroke risk assessment and further stroke research. In addition, new biomarkers for the genetic causes of stroke and new targets for gene therapy are on the horizon. Conclusion: We summarize the latest evidence and perspectives of ischemic stroke genetics that may be of interest to the physician and useful for day-to-day clinical work in terms of both prevention and treatment of ischemic stroke.

## 1. Background

Stroke is defined as a lack of blood flow in the brain that can cause neurological deficits [1,2,3]. The major cause of ischemic stroke is arterial atherosclerosis [4,5]. Other cause of stroke is genetic etiology [6,7] and about 15% of strokes are observed in people aged 18–49 years old [8]. Monogenic and polygenic disorders represent about 7% and 38%, respectively, of all stroke causes [9,10,11,12,13,14,15,16,17,18,19,20,21]. Different risk scores for stroke outcomes are described in the literature, such as the Genetic Risk Score (GRS) and the Extended Polygenic Scores (PRS) [22,23]. Different classifications are reported in relation to genetic risk factors and previous studies have identified 10 loci associated with stroke [23].

## 2. Materials and Methods

In this narrative review, we included clinical trials published by Pubmed until 28 February 2022. The keywords used were ischemic stroke, genetic, causes, epigenetic, etiopathogenesis. All paper and clinical trials published by Pubmed were studied by two authors. We excluded studies written in languages other than English. Three authors (P.C., T.C. and E.M.) reviewed all articles and all studies were qualitatively analyzed.

## 3. Results

### 3.1. Association between Genetic Alterations and Risk Factors

Based on linkage analysis, a better understanding was gained for the inheritance of stroke only for those forms of ischemia linked to monogenic disorders. However, it is known that the genetic causes of stroke are also polygenic.

#### 3.1.1. Monogenic Alterations

Monogenic alterations are responsible for about 1% of strokes those in the young population. Other studies reported that they are responsible for 7% of strokes [9,10,11,12,13,14,15,16,17,18,19,20,21]. We reported the main forms of monogenic stroke:A.CADASIL—It is an autosomal-dominant arteriopathy characterized by subcortical infarct and leuko-encephalopathy (CADASIL). It is a major monogenic cause of ischemic stroke [24] and it is related to a pathogenic variant of the NOTCH3 gene [25,26] that is inherited in an autosomal-dominant way. The characteristic of CADASIL is deposition of granular osmiophilic material (GOM) in the vascular wall [27].B.CARASIL—It is an autosomal-recessive cerebral artery disease, and it is related to a mutation of the HTRA1 gene [28]. It is defined by intense arteriosclerosis without amyloid deposits and intimal proliferation or loss of smooth muscle cells in small arteries [29]. Retinal vasculopathy with systemic manifestations (RVCL-S) affects the small vessels of the retina, brain, kidneys and liver and it has an autosomal-dominant inheritance pattern [30]. The gene responsible for RVCL-S is TREX1 and it encodes the TREX1 protein, which, after mutation, makes endothelial cells more vulnerable to oxidative DNA damage [31]. TREX1 is detectable in the microglia surrounding the white matter micro vascularization, and probably the incorrect localization of TREX1 may induce the vulnerability to white matter failure [32,33]. It has been reported that these circulating endothelial markers are present in retinal vasculopathy with cerebral leukoencephalopathy [34,35].C.MELAS—It is a mitochondrial encephalomyopathy characterized by lactic acidosis and stroke-like episodes (MELAS). The genetic alterations typical of this disease involve the bioenergetic functions of the cell [36]. MELAS is associated with:
growth retardationlactic acidosisneuromyopathyepilepsymigraine-like headacherecurrent stroke-like episodes (SLE) such as ischemic stroke [37,38,39,40]. Recent data [37,38,39,40] indicate that neuronal and/or glial damage is caused by cerebrovascular angiopathy. In this condition, symmetrical calcifications in the basal ganglia are reported [37,38,39,40].D.Sickle cell anemia—It is caused by a point mutation (GAG to GTG) in the ß-globin gene [41]. The clinical manifestations are characterized by chronic hemolysis and acute vaso-occlusive crisis. The prevalence of stroke is about 3.75%, specifically in the first decade of life [42]. Large-vessel occlusion and high development of aneurysms are reported [43,44,45,46].E.Type IV Ehlers–Danlos syndrome (EDS-IV)—It is a hereditary disorder associated with an abnormal synthesis of procollagen III. It is characterized by heterozygous mutations in the COL3A1 gene [47], and EDS-IV is characterized by facial malformations, skin changes, a tendency to develop bruises and hematomas, and arterial, digestive, and obstetric complications. In this condition, intracranial aneurysms, dissection of the vertebral and carotid arteries and spontaneous rupture of large and medium-sized arteries are reported [48,49].F.Homocystinuria—It is an autosomal-recessive disease affecting the metabolism of the amino acid methionine. Prevalence has been estimated at 1 in 344,000 [50]. It is characterized by an abnormal accumulation of homocysteine and its metabolites in the blood and urine. Clinical manifestations are heterogeneous and include abnormalities of the eye, skeleton, and nervous system. The complications are related to endothelial damage and the stimulation of platelet aggregation [51].G.Elastic pseudoxanthoma—It is known as Groenblad–Strandberg syndrome, and it is an autosomal-recessive elastic tissue disease caused by mutations in the ABCC6 gene [52]. Cardiovascular conditions are prevalent in females, and in affected subjects there is an accelerated arteriosclerosis mainly affecting small and medium-sized arteries [53].H.Fabry disease—It is X-linked recessive and caused by a mutation in the GLA gene encoding the enzyme alpha-galactosidase A [54,55]. Neurological manifestations include polyneuropathy, autonomic dysfunction, and brain manifestations. Stroke may be the first manifestation of the disease, which more frequently affects the posterior cerebral circulation.I.Marfan syndrome—It is an autosomal dominant condition of the connective tissue, associated with mutations in the fibrillin-1 gene. The incidence is estimated at 2–3/10,000. The clinical characteristics include cardiovascular, skeletal, and ocular symptoms [56,57].L.Type IV collagen dysfunctions α1 and α2—Both are autosomal dominant conditions, caused by mutations in the COL4A1 (13q34) and COL4A2 (13q34) genes. The clinical features include neurological features (such as stroke, migraine, infantile hemiparesis, epilepsy) and systemic symptoms [58]. Forms of COL4A1 mutations include infantile hemiparesis, seizures, migraine with aura, single or recurrent intracerebral hemorrhages, eye symptoms and muscle spasms [58].M.Cerebral cavernous malformations—These are known as cavernous angiomas or cavernomas. The prevalence is 0.8% in the general population. The familial form is autosomal dominant and associated with KRIT1 mutations. The clinical manifestations are characterized by seizures, headaches, and intracranial hemorrhage [59]. Cavernous brain malformations can be familiar or sporadic [60].N.Cerebral amyloid angiopathy—It is responsible for 15% of hemorrhagic strokes. The hereditary form is rare. The degenerative process leads to the development of microaneurysms as well as to hemorrhagic and ischemic lesions of the brain [61].

#### 3.1.2. Polygenic Cerebrovascular Diseases

Polygenic cerebrovascular diseases are caused by multiple genes. It has been reported that he 38% of variability observed for the thickness of the common carotid artery is attributed to genetic background [62]. Different studies [14,63,64,65] highlight the crucial characteristics of stroke genomics:The largest genetic correlation was found for arterial hypertension.This suggests a strong link with a cardiac mechanism.New stroke risk can constitute drug targets for antithrombotic therapy.

Different risk scores for stroke outcome are described in the literature, such as the Genetic Risk Score (GRS) and the Extended Polygenic Scores (PRS) [22,23]. Different classifications are reported in relation to genetic risk factors, and previous studies have identified 10 loci associated with stroke [23].

### 3.2. Atherosclerotic Stroke

It has been reported that 38% of the variability observed for the thickness of the common carotid artery is attributed to genetic background [17].

#### 3.2.1. Atherosclerotic Stroke

Ischemic lesions of small diameter (<5 mm), often multiple, with localization in the distribution territory of the arterioles penetrating the basal ganglia, pons, internal capsule, and white matter are called micro lacunar strokes. These strokes are early onset. There are two subtypes of lacunar stroke:isolated lacunar stroke.multiple lacunar stroke with leuko-araiosis.

Beyond the risk factors, numerous studies have revealed several genetic mechanisms that increase the risk of lacunar stroke [19,21,66,67,68,69]:altered oxidative phosphorylation pathways.various single nucleotide polymorphisms.

A genetic polymorphism was found at the level of a locus on the chromosome located on the long arm of chromosome 6 (6p25) [70]. Proteins encoded by the FOXF2 gene play a crucial role in DNA repair, cell proliferation and organ development. The associations of this locus with the expression of ZCCHC14 and DNA methylation suggest that together they contribute to the etiopathogenesis of stroke by altering the function of the regulatory elements of cell proliferation [71,72]. Other authors have found several microlacunar stroke–related polymorphisms involving the TRIM65 and TRIM47 genes encoding ubiquitin-like proteins capable of regulating a range of intracellular events [73]. Since these are mainly early-onset strokes, the same study showed that higher blood pressure levels were found in subjects with polymorphisms of these genes, while the polymorphisms of other genes, CSN3, HLA-DPB1 and SH3TC1, are associated with cardiovascular diseases and diabetes mellitus [74,75].

#### 3.2.2. Cardioembolic Stroke

So far, no genes have been detected that significantly link atrial fibrillation to stroke. However, in ischemic stroke, two genes (PITX2 and ZFHX3), located in the short arms of chromosomes 4 (4q25) and 16 (16q22), respectively, have been reported as significant risk factors for stroke [13,68]. The PITX2 gene encodes a protein that regulates the expression of the procollagen lysyl hydroxylase gene, which is required for the production and stabilization of vessel-supporting collagen. The ZFHX3 gene encodes a protein that regulates myogenic and neuronal differentiation. A single-nucleotide polymorphism in the PITX2 and ZFHX3 genes and two other significantly associated genes, ZNF566 and PDZK1IP1, increase the risk of stroke [20,69].

#### 3.2.3. Cases of Stroke Related to Other Diseases

The authors of the EuroCLOT study found that the genes of the ABO system are associated with the Von Willebrand factor and factor VIII levels, and therefore, if present in a polymorphic form, they are related, with a higher prevalence, to microlacunar ischemic strokes [66]. Additionally, the HLA system and the major histocompatibility complex, especially in the inflammatory response mediated by natural killer cells, is correlated with the risk of stroke, and systemic inflammatory states have also been linked to the main autoimmune diseases [68].

### 3.3. Epigenetic Causes of Stroke

Stroke is a multifactorial disease [76]. Numerous studies [19,21,66,67,68,69] have described the complex genetic and molecular mechanisms involved in the pathogenesis of ischemic stroke (IS). Epigenetic mechanisms represent new factors involved in the occurrence of stroke [76].

#### 3.3.1. DNA Methylation

Epigenetic modifications, unlike gene sequence changes, are reversible [77,78]. DNA methylation modulates the interaction between transcription factors with their specific binding sites [79,80]. Pharmacological inhibition of DNA methyltransferase (DNMT) has been shown to decrease the methylation of DNA and to reduce ischemic brain damage in the rat model with middle cerebral artery occlusion [80]. DNA hypermethylation is now associated with many pathogenetic mechanisms involved in ischemic brain damage, reported as follows:Hypermethylation of the genetic sequences encoding thrombospondin has been observed in vitro as a cause of ischemia. This factor is involved in platelet aggregation as well as in the neo-angiogenesis, for example, linked to post ischemic damage [80].Elevated levels of both triglycerides and low-density lipoprotein (LDL) cholesterol are associated with an increased risk of stroke [81,82,83,84,85]. This phenomenon is correlated in the human organism with the role of apolipoprotein E, a lipoprotein involved in lipid metabolism [86]. It has been observed that polymorphisms of the ApoE gene, the protagonist of the expression of this lipoprotein, are correlated with a greater progress of atherosclerosis [86]. ApoE hypermethylation can be prevented by reducing homocysteine levels [87].In humans, it is also known that high plasma homocysteine levels are an independent risk factor for atherosclerosis and coronary heart disease [88,89,90]. Homocysteine is an amino acid present in the body in very small quantities; it derives from biochemical reactions that start from methionine, an amino acid that we ingest by consuming foods such as meat, eggs, legumes, and dairy products. Alterations in homocysteine metabolism are associated with a higher incidence of stroke. Homocysteine metabolism is regulated by dietary factors such as methionine, B vitamins, methylenetetrahydrofolate reductase (MTHFR), cystathionine beta-synthase (CBS) and methionine synthase (MS). Above all, the hypermethylation of CBS leads to low enzymatic activity and, therefore, the accumulation of plasma homocysteine [90]. High levels of plasma homocysteine are also associated with DNA hypermethylation of thrombomodulin in patients with ischemic stroke [89,90,91].The DNA hypermethylation process also seems to be at the basis of the malfunction of the cyclin-dependent kinase inhibitor 2B, a gene involved in the pathogenesis of calcium deposition in the arterial wall [89,90,91].Baccarelli et al. [91] reported that hypomethylation of long nucleotide elements was associated with increased levels of a vascular cell adhesion protein (VCAM-1). The association between hypomethylation and the expression of VCAM-1 may be a link for cardiovascular and cerebrovascular diseases [92,93]. In discussing the role of epigenetic factors involving inflammation on the pathogenesis of ischemic injury, it should be remembered that the hypomethylation of TNF receptor–associated factor 3 (TRAF3) and protein phosphatase 1A (PPM1A) was associated with increased risk for stroke in patients treated with antiplatelet therapy [94,95].

#### 3.3.2. Histone Modifications

Another epigenetic factor connected to the pathogenesis of ischemic brain damage is represented by the histone modifications that play a fundamental role in the spatial organization of DNA. Histone–DNA interactions affect chromatin structure, thereby directing the accessibility of transcriptional regulators to DNA-binding elements [96,97]. Histone methylation can activate or inhibit gene expression [98]. Studies have highlighted the role of histone modifications in stroke. Low HDAC levels were associated with impaired cell proliferation and angiogenesis [99] and with loss of blood–brain barrier integrity [99,100,101,102]. The inhibition of HDCA is still not a viable therapeutic path today due to the low bioavailability of these drugs in the brain and to the large number of side effects that do not justify their routine use in the face of such modest although significantly valid results [103,104,105,106,107].

#### 3.3.3. Non-Coding RNA

The regulatory mechanisms attributable to RNA are counted among the epigenetic factors of stroke. The regulation of gene expression involves microRNAs (miRNAs). MiRNAs are described in cell proliferation and differentiation, apoptosis, fat metabolism and hematopoiesis [108]. LncRNAs are a regulator of vascular function, and their dysregulation contributes to cardiovascular disease [109,110]. Studies have revealed alterations in miRNA levels in arterial hypertension, dyslipidemia, atherosclerosis, and inflammation [111,112,113,114]. Furthermore, changes in miRNA levels have been identified at different stages of stroke, and it is possible that they can be used for diagnostic, prognostic, and therapeutic purposes [115]. From studies, miRNAs have been shown to influence the function of genes that induce inflammatory pathways, such as interleukin 1β and 6, intercellular adhesion molecule 1 (ICAM1) and cyclooxygenase 2 (COX2) [116,117,118,119,120,121,122,123]. These processes play a crucial role during post-ischemic brain remodeling [124,125]. In rats, a reduced level of miR-122 is related to reduced endothelial integrity, apoptosis, and inflammation [126]. These findings describe the growing role of miRNAs in the pathogenesis of stroke [127,128,129,130].LncRNAs ANRIL (non-coding antisense RNA in the INK4 locus), MALAT1 (metastasis-associated lung adenocarcinoma transcript 1), MEG3 (maternally expressed 3), TUG1 (taurine upregulated 1) [131] and ANRIL expression were significantly increased in stroke models [131]. MALAT1, MEG3 and TUG1 are involved with LncRNAs in the regulation of cell proliferation, inflammation, and apoptosis [132]. These results suggest that LncRNAs can have a crucial role in the development of ischemic stroke [133,134,135,136,137].

## 4. Discussion

The purpose of this narrative review was to better clarify the current advances in the study of the genesis of the various forms of stroke. Today, it is possible to state that the application of genetics to our understanding of stroke is an opportunity that brings new insights to the knowledge of pathogenesis and, therefore, to any therapeutic targets, combined with the possibility of containing costs to relatively modest figures compared to the past. This allows doctors to have greater possibilities to prevent and treat stroke and to classify these pathological forms into different subtypes based on their etiologies and genetic risks. The development of corresponding monoclonal antibodies could represent a new form of personalized stroke care. The use of genetic risk scores is a genomic prediction method that can be calculated by evaluating information on multiple genetic markers and variants. Numerous genetic risk scores have already been proposed for stroke [138]. Malik et al. [20] found a statistically significant correlation for the risk of ischemic stroke. Similar results have been reported regarding the relationship between genetic risk factors for hypertension and the risk of ischemic stroke, although in this case, the results were also similar in predicting stroke with respect to the presence of the risk factors themselves [139,140,141,142]. Despite the formulation of several genetic risk panels, it can be noted that in most cases, the predictive value is relative or comparable to that of the same environmental risk factors. This problem could be overcome if future studies considered the use of genetic risk panels to stratify the risk for each specific stroke subtype. The use of a panel including tests for 32 individual genetic polymorphisms proved to be more effective in predicting the risk of ischemic stroke in a population of patients with atrial fibrillation. Another potential benefit derived from the knowledge of the genetic phenomena of stroke concerns the development of potential treatments aimed at specific therapeutic targets. It has recently been seen that ischemic stroke reduces gene expression due to the suppression of the acetylation of histones H3 and H4 [143,144]. Histone deacetylase inhibitors induce neurogenesis and angiogenesis in damaged brain areas by promoting functional recovery after cerebral ischemia. In the CADASIL form, it has been reported that the pathogenesis of stroke is linked to the accumulation of characteristic extracellular deposits. However, they protect the vascular endothelium from alterations in cerebral blood flow [145,146,147,148,149]. Ultimately, the knowledge of stroke-risk loci increases the possibility of having new drug targets for antithrombotic therapy, thus highlighting the potential of stroke genetics in the field of drug discovery, and laying the foundations for an understanding, in a more concrete way, of the intimate relationship that exists between the genetic characteristics of each individual and stroke.


**The Key Messages**


Gene expression is involved in brain inflammation and remodeling after stroke.Epigenetic mechanisms may induce different injury responses to cerebral ischemia.Epigenetics has recently sparked interest because it appears to offer an alternative therapeutic solution by altering our genetic “destiny”.

## 5. Conclusions

The knowledge of stroke-risk loci increases the possibility of having new drug targets for antithrombotic therapy, thus highlighting the potential of stroke genetics in the field of drug discovery, and laying the foundations for an understanding, in a more concrete way, of the intimate relationship that exists between the genetic characteristics of each individual and stroke. This knowledge can help in the acute treatment or prevention of stroke.

## Data Availability

Not applicable.

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
