# Peer review of "Stroke and Etiopathogenesis: What Is Known?"

_genes, 2022, doi:10.3390/genes13060978_

Round 1

Reviewer 1 Report

In this original article entitle “Stroke and epigenetic: what is known?”, Ciarambino et al. presents a Review up to date about monogenic diseases that can cause stroke, polygenic diseases related with stroke, genes associated with stroke subtypes, and epigenetics markers associated with stroke.

However, after reading this review I consider that the title of this article does not fit with the content of it. The article reviews the knowledge on stroke genetics, providing a summary of the molecular function of each one of the genes associated with stroke. Thus, as it is not only focus on Epigenetics, I recommend to change the title in order to fit better the content of this review article.

Additionally, there some other issues that I would like to report:

  1. Line 28: I do not agree with this definition of stroke. Stroke is the lack of blood flow in the brain that can cause neurological deficit, but not the neurological deficit itself.
  2. Line 32-36: Re-write this paragraph. It is not well explained.
  3. Unify format of section 3.1.
  4. Line 133: I also disagree with this phrase. Stroke is a polygenic disease itself. Additionally, the references cited here does not support the statement made by the authors. Reference 67 find out that the 38% of variability observed for the thickness of the common carotid artery was attributed to the genetic background. This cannot be directly extrapolated to stroke incidence.
  5. Line 146: I have not found in any of the references added that the incidence of stroke increases in a 39% due to the presence of SNPs in this locus. References 25 contains the Odd ratio found for this locus that is around 1.29, which means an increase of stroke risk (not incidence) of 29%.
  6. Line 163: There is a “(” lacking the close parenthesis “)” that does not contain anything. Please review if it is a typo or if there is something missing.
  7. Line 177: the title is in the wrong positon.
  8. Line 180: they should clarify that are risk factors for ischemic The actual phrase can be confused for risk factors of all-types of stroke (hemorrhagic and ischemic).
  9. Section 3.1.5. Epigenetic cause of stroke: Adding paragraph separations and subsections will help with the reading of this section.
  10. Line 195-196. This phrase lacks a verb. “DNA methylation [verb: modulates, regulates, inhibits?] the interaction between transcription factors with their specific binding 195 sites (88).”
  11. Line 298-299: I do not see in the prior text any references to neuronal signal and epigenetics. I would remove this key message.
  12. Line 303: there is a typo. “Epigenetic therapy can be a potential way that influence stroke care pathways”

Taking all into account, I considered that this manuscript needs to be revised and its quality improved in order to be considered for publication in Genes.

Author Response

Naples, 8 May 2022

Dear Editor and Reviewers,

            Please, find enclosed the revised version of the manuscript entitled “Stroke and ethiopathogenis: what is known?2. We thank the Editor and the Reviewers for their comments and we hope that the following changes will now make the manuscript suitable for publication on the Genes. Please see the following list of the underlined changes made in manuscript.

Reviewer 1 comment’s:

  1. Line 28: In agreement with the Reviewer’s comments we now reported this definition: “Stroke is the lack of blood flow in the brain that can cause neurological deficit”.
  2. Line 32-36: In agreement with the Reviewer’s comments we now re-write this paragraph.
  3. In agreement with the Reviewer’s comments we now unify format of section 3.1.
  4. Line 133: In agreement with the Reviewer’s comments we now reported as following. “It has been reported that he 38% of variability observed for the thickness of the common carotid artery was attributed to the genetic background.”
  5. Line 146: In agreement with the Reviewer’s comments we now reported as following: “in an increase of stroke risk (not incidence) of 29%.”
  6. Line 163: In agreement with the Reviewer’s comments we now have eliminated “(”
  7. Line 177: In agreement with the Reviewer’s comments we now reported the new title” Atrial fibrillation and stroke. “.
  8. Line 180: In agreement with the Reviewer’s comments, we now reported as following: “in ischemic stroke”.
  9. Section 3.1.5: now 3.2.3. In agreement with the Reviewer’s comments we now report the reading of this section.
  10. Line 195-196. In agreement with the Reviewer’s comments we now report the verb as following sentence “DNA methylation modulates, the interaction between transcription factors with their specific binding sites (88).”
  11. Line 298-299: In agreement with the Reviewer’s comments we now remove this key message.
  12. Line 303: : In agreement with the Reviewer’s comments we now correct as following sentence:” “Epigenetic therapy can be a potential way that influence stroke care pathways”.
  13. In agreement with the Reviewer’s comment the English language is revised by native American speaker.

Best regards,

Tiziana Ciarambino

MD, PhD

Reviewer 2 Report

Stroke is defined as neurological deficit due to cerebral infarction or intracerebral 28 haemorrhage (1-3).

3 refferences for 15 word sentance might be too much.

It has been reported that it is related to cognitive impairment (4-6)

If you mention this, then you have to be more specific, and you dont talk about cognitive inpariment after that sentance. So I suggest to remove the sentence at. 

Monogenic disorders represent about 7% of stroke causes, while polygenic disorders 37 represent 38% of stroke causes. Genetic risk score (GRS) and Extended Polygenic Scores 38 (PRS) describe the contribution to a specific stroke outcome (12-14). Previous studies have 39 identified 10 loci associated with stroke (15-24) and different classification are reported in 40 relation to genetic risk factor (25-26). In particular, Illinca et al. (26) reported that others 41 risk are related to previous stroke condition (as hypertension, diabetes, etc). 42

Here also, you need to be more specific. 

The background should be rewritten.

Conclusions 305 The knowledge of stroke risk loci increases the possibility of having new drug targets 306 for antithrombotic therapy, thus highlighting the potential of stroke genetics in the field 307 of drug discovery and laying the foundations for an understanding in a more concrete 308 way of the intimate relationship existing between the genetic characteristics of each indi- 309 vidual and stroke. 

Do you mean for acute treatment or prevention goals.

Author Response

                                                                                                          Naples, 8 May 2022

Dear Editor and Reviewers,

            Please, find enclosed the revised version of the manuscript entitled “Stroke and ethiopathogenis: what is known?2. We thank the Editor and the Reviewers for their comments and we hope that the following changes will now make the manuscript suitable for publication on the Genes. Please see the following list of the underlined changes made in manuscript.

Reviewer 2 comment’s:

1.In agreement with the Reviewer comments we now report the following sentence: “Stroke is defined as the lack of blood flow in the brain that can cause neurological deficit,…..”

  1. In agreement with the Reviewer comments, we now report the following sentence: “In particular, the dementia is associated to vascular ischemic brain damage (4-6).”
  2. In agreement with the Reviewer comments, we now rewrite the background as following sentenceStroke is defined as the lack of blood flow in the brain that can cause neurological deficit (1-3). It has been reported that it is related to cognitive impairment (4-6). In particular, the dementia is associated to vascular ischemic brain damage (4-6). The major cause for ischemic stroke is arterial atherosclerosis (7-8) and about 15% of stroke observed in people aged 18-49 years old (11). Other causes are reported in literature. In particular, monogenic and polygenic disorders represent about 7% and 38%, respectively, of all stroke causes. Different risk scores for stroke outcome are described in literature, such as Genetic risk score (GRS) and Extended Polygenic Scores (PRS) (12-14). Different classifications are reported in relation to genetic risk factor and previous studies have identified 10 loci associated with stroke (15-26). Illinca et al. (26) reported that other risks are related to previous stroke condition (as hypertension, diabetes, etc).”

  1. In agreement with the Reviewer comment’s we now reported the following sentenced “This knowledge can help the acute treatment or prevention for stroke”.

Best regards,

Tiziana Ciarambino

MD, PhD

Round 2

Reviewer 1 Report

After reviewing the edited manuscript of Ciarambino et al., I considered that its quality has been improve. Despite that, I have notice some issues:

  1. Line 39-44: Some words are in a different text font.
  2. Line 137: There is a typo: “It has been reported that the 38% of variability observed for the thickness of the common carotid artery was attributed to the genetic background (67).” Additionally, I considered that this phrase should appear in the subsection “Atherosclerotic stroke”.
  3. Subsections of section 3 are unclear and difficult to follow. Please review them. Using the text as guidance I suggest to divide it into three parts: 3.1. Monogenic alterations, 3.2. Polygenic cerebrovascular diseases, 3.3. Epigenetic causes of stroke.

Then section 3.2 can be divided into: 3.2.1. Atherosclerotic stroke, 3.2.2. Cardioembolic stroke, 3.2.3. Cases of stroke related to other diseases.

Finally, section 3.3 could be divided into: 3.3.1. DNA methylation, 3.3.2. Histone modifications, 3.3.3 non-coding RNA.

  1. Regarding open paragraph of section “3.2 Polygenic cerebrovascular diseases”, the authors could include here some of the work about stroke heritability and the estimate percentage described until know by the found gene associations (this information is in reference 12), and may be introduce the genetic scores as they are discussed later in the manuscript.
  2. Line 314-315: “Epigenetic processes are essential for complex brain functions, allowing proper neuronal signal.” This key message has not been removed. As I mention before, the text does not show any evidence for this key message. I suggest to remove this key message.

Taking all into account, I consider that this manuscript needs further polishing before been accepted for publication in Genes.

Author Response

Naples, 13 May 2022

Dear Editor and Reviewers,

            Please, find enclosed the revised version of the manuscript entitled “Stroke and ethiopathogenis: what is known? We thank the Editor and the Reviewers for their comments and we hope that the following changes will now make the manuscript suitable for publication on the Genes. Please see the following list of the underlined changes made in manuscript.

Reviewer 1 comments

  1. Line 39-44: In agreement with the Reviewer’s comments we now correct the text font.
  2. Line 137: In agreement with the Reviewer’s comments we now report the phrases in the subsection “Atherosclerotic stroke”.
  3. Subsections of section 3: In agreement with the Reviewer’s comments we now divide it into three parts: 3.1. Monogenic alterations, 3.2. Polygenic cerebrovascular diseases, 3.3. Epigenetic causes of stroke.
  4. Section 3.2: In agreement with the Reviewer’s comments, we now divide it into three parts: 3.2.1. Atherosclerotic stroke, 3.2.2. Cardioembolic stroke, 3.2.3. Cases of stroke related to other diseases.
  5. Section 3.3 In agreement with the Reviewer’s comments, we now divide it into: 3.3.1. DNA methylation, 3.3.2. Histone modifications, 3.3.3 non-coding RNA.
  6. Paragraph of section “3.2 Polygenic cerebrovascular diseases”, we now report it.
  7. Line 314-315: In agreement with the Reviewer’s comments we now remove this message.

Best regards,

Tiziana Ciarambino

MD, PhD

Reviewer 2 Report

Now the bacground section is even more unclear. 

It has been reported that it is related to cognitive impairment (4-6). In particular, 35
the dementia is associated to vascular ischemic brain damage (4-6).

These sentences are now without any context. there is still the same amout of refferences with no perpouse. 11 lines and 26 refferences!

In materials and methods:

it is not clear what was studied by the two authors. 

In key messages

Epigenetic therapy can influence techniques to treat ischemic stroke. 

Epigenetic therapy can be a potential way that influence stroke care pathways 

Is not supported by the results or discusstion. 

Author Response

XCV

Naples, 13 May 2022

Dear Editor and Reviewers,

            Please, find enclosed the revised version of the manuscript entitled “Stroke and ethiopathogenis: what is known? We thank the Editor and the Reviewers for their comments and we hope that the following changes will now make the manuscript suitable for publication on the Genes. Please see the following list of the underlined changes made in manuscript.

Reviewer 2  comments

  1. In agreement with the Reviewer comments, we now remove the sentence….It has been reported that it is related to cognitive impairment (4-6). In particular, the dementia is associated to vascular ischemic brain damage (4-6).
  2. In agreement with the Reviewer comments, we now describe in materials and methods paper and clinical trials published on the stroke were studied by two authors.”
  3. In agreement with the Reviewer comments, we in key messages we remove the phrases: Epigenetic therapy can influence techniques to treat ischemic stroke. Epigenetic therapy can be a potential way that influence stroke care pathways”.

Best regards,

Tiziana Ciarambino

MD, PhD
